# DeepEST: A Python Library for Spatio-Temporal Epidemiology Prediction

### Yuhang Wang
Peking University
Beijing, China
2301210413@stu.pku.edu.cn

### Yinghao Zhu
Peking University
Beijing, China
yhzhu99@gmail.com

### Lifang Liang
Peking University
Beijing, China
20374090@buaa.edu.cn

### Yasha Wang
Peking University
Beijing, China
wangyasha@pku.edu.cn

### Ewen M. Harrison
University of Edinburgh
Edinburgh, Scotland, UK
ewen.harrison@ed.ac.uk

### Liantao Ma
Peking University
Beijing, China
malt@pku.edu.cn

### Junyi Gao*
University of Edinburgh
Health Data Research UK
Edinburgh, Scotland, UK
junyi.gao@ed.ac.uk

## ABSTRACT

The COVID-19 pandemic highlighted the importance of spatio-temporal epidemiology prediction, which utilizes both spatial and temporal data to forecast disease dynamics. This approach is critical for public health management, particularly during pandemics, due to its ability to analyze disease spread patterns over time and across different regions. Traditional data sources provide extensive spatio-temporal datasets, yet they underscore a need for sophisticated predictive tools that are accessible to public health researchers without deep technical expertise. To address this, we introduce DeepEST, a Python library designed to facilitate deep learning-based spatio-temporal epidemiology prediction. DeepEST integrates advanced predictive modeling techniques such as graph neural networks and recurrent neural networks with traditional epidemiological models, offering a comprehensive suite of tools for data preprocessing, model development, and visualization. This library simplifies the development and application of predictive models, reducing the technical barriers for researchers and enhancing the capability of the public health community to effectively respond to epidemiological threats. This paper details the development and capabilities of DeepEST, showcasing its potential to democratize epidemiological research and public health response strategies. Our code is available at https://github.com/v1xerunt/DeepEST.

## 1 INTRODUCTION

The COVID-19 pandemic has underscored the critical importance of accurate predictive modeling in managing public health crises. Spatio-temporal epidemiology prediction, which leverages both spatial and temporal data, has emerged as an important tool in understanding and combating pandemics and addressing public health challenges. Particularly, deep learning-based spatio-temporal prediction models have demonstrated significant potential due to their ability to extract complex spatial and temporal interdependencies. By analyzing the patterns of disease spread over time and across geographies, these models offer invaluable insights into the dynamics of epidemics, helping public health officials make informed decisions about interventions and resource allocation [12, 13, 20].

Spatio-temporal epidemiology prediction requires building predictive models to extract and utilize spatial and temporal patterns

from large-scale multi-location time series. Using the COVID-19 pandemic as an example, from the data perspective, several initiatives have been launched to aggregate and organize data during the pandemic, including the US COVID-19 forecast hub [7], the National COVID Collaborative (N3C) [14], and the Johns Hopkins University (JHU) COVID-19 Data Platform [10]. These platforms and data hubs have provided structured, accessible population-level or individual-level data for healthcare research and analysis. However, there remains a conspicuous gap in the availability of specialized tools that simplify the use and development of predictive models to fully utilize these data in public health research.

Utilizing or developing spatio-temporal pandemic prediction models remains a formidable challenge. Constructing these prediction models requires complicated preprocessing of spatio-temporal data and location connectivity construction. The complexity of state-of-the-art spatio-temporal prediction models, which often involve advanced methodologies such as graph neural networks, cross-location attentions, and recurrent neural networks, necessitates a high level of expertise in both deep learning and epidemiological modeling. These issues are creating substantial barriers for public health researchers.

To bridge this gap, we introduce DeepEST, a comprehensive Python library designed specifically for deep learning-based **S**patio-**T**emporal **E**pidemiology prediction. DeepEST is the first Python package that provides data preprocessing, modeling, evaluation, and visualization tools for spatio-temporal data, especially for regional pandemic time series. DeepEST democratizes access to advanced predictive modeling techniques by integrating deep learning, machine learning, and traditional epidemiology compartmental models under a unified framework, enabling users to leverage state-of-the-art methodologies without the need for extensive technical expertise. The library's model zoo is comprehensive, with both deep learning and machine learning, temporal and spatio-temporal prediction models, enabling users to choose the most appropriate model for different prediction tasks and settings. By simplifying the process of model application and reducing the entry barriers for researchers, DeepEST aims to enhance the public health community's ability to forecast and respond to epidemiological threats effectively. This paper outlines the development of DeepEST, its

---

*Corresponding author.

key features, and the potential it holds for transforming the landscape of epidemiological research in the face of ongoing and future pandemics.

## 2 RELATED WORK

In the domain of spatio-temporal data analysis, there are some existing tools and libraries for developing spatio-temporal prediction models. Torch Spatiotemporal (TSL) [5] is a Python library for PyTorch that facilitates the processing of spatio-temporal data, with a focus on Graph Neural Networks (GNNs). It offers utilities for data preprocessing and model prototyping, simplifying the workflow. The library is designed to handle applications like environmental monitoring, urban computing and traffic analysis. Pytorch Geometric Temporal(PGT) [18] extends the capabilities of PyTorch Geometric, focusing on the analysis and prediction of dynamic graph-structured data with temporal sequences. Its main contribution is the integration of a large number of general dynamic spatio-temporal models.

However, both Torch Spatiotemporal (TSL) and PyTorch Geometric Temporal are not designed for epidemiology and public health research. They lack built-in pre-processing tools for common epidemiology data. Neither of them supports epidemiological models and also lacks of evaluation pipelines and visualization tools, which are common use cases in public health studies. The comparison of Torch Spatiotemporal, Pytorch Geometric Temporal and our toolkit DeepEST is shown in Table 1.

**Table 1: Comparison of key features in Torch Spatiotemporal (TSL), Pytorch-Geometric-Temporal (PGT), and DeepEST.**

| Feature | TSL | PGT | DeepEST |
|---|---|---|---|
| **Data source and pre-processing** | | | |
| Temporal granularity aggregation | – | – | ✓ |
| Sliding window processing | ✓ | – | ✓ |
| Adaptive graph construction | – | – | ✓ |
| Spatio-temporal data split | ✓ | ✓ | ✓ |
| External dataset support | – | ✓ | ✓ |
| **Modeling** | | | |
| Statistical epidemiology models | – | – | ✓ |
| General machine learning models | – | – | ✓ |
| Deep-based time series models | – | – | ✓ |
| Spatio-temporal epidemiology models | – | – | ✓ |
| **Post-processing** | | | |
| Evaluation pipeline | ✓ | – | ✓ |
| Graph and prediction visualization | – | – | ✓ |

## 3 DEEPEST

DeepEST is developed and executed within a Python 3 environment, leveraging libraries including numpy, pandas, matplotlib, scikit-learn, and Pytorch. The key modules and pipelines of using DeepEST are shown in Figure 1.

### 3.1 Spatio-temporal Data Preprocessing

The spatio-temporal data preprocessing module is at the core of the DeepEST library, offering data cleansing, transformation, and normalization essential for public health prediction tasks. This module adeptly handles spatio-temporal data, with enhanced functionalities tailored for Graph Neural Networks (GNNs):

- **Temporal aggregation:** This module facilitates data aggregation features according to diverse temporal scale requirements, suitable for periodic analyses like weekly or monthly cycles.
- **Spatio-temporal data split:** This module enables dynamic segmentation of data across temporal and spatial dimensions, structuring datasets for model training into training, validation, and testing sets.
  - \* In the time dimension, users can divide the training set, validation set, and test set according to the time sequence by proportion;
  - \* In the spatial dimension, the user can specify the target region of the training set, validation set, and test set through the index of this region in the data.
  By allowing users to split data across both spatial and temporal dimensions, DeepEST offers simulations of evaluating model performance on new unseen time periods or locations, which are common evaluation settings in public health research.
- **Sliding window processing:** This module allows users to define input and forecasting windows for time series predictive models, enabling data serialization through a sliding window approach. More specifically, as is shown in Figure 2, the user can specify the prediction window and the output window, as well as the time interval between them (as specified by the delay parameter). The moving step of the window can be specified with the STEP parameter.
- **Spatio-temporal data normalization:** This module employs normalization techniques including z-score standardization and min-max scaling to ensure the stability and consistency of the input data for predictive models.

A key feature in DeepEST is adaptive graph construction using location-wise static data, such as demographics and public health resources. This feature allows users not to have to provide a predefined location connectivity graph. DeepEST calculates the distance (e.g., L1 distance, L2 distance, haversine distance, or custom distance functions) between locations based on their static data, and then builds edges between pairs of nodes. Moreover, as is shown in Figure 4, DeepEST also provides a visualization tool that makes it easier for users to see connections between individual nodes.

### 3.2 Model Zoo

DeepEST support both machine learning and deep learning, temporal and spatio-temporal models.

- **Spatio-temporal data prediction models**: This type of model focuses on analyzing and predicting the spread and correlation of epidemics across different regions. They are able to integrate spatio-temporal factors and time series data to reveal patterns and trends in disease transmission. Through advanced algorithms, we can predict the development of disease in each region of the test set over a period of time in the future, providing a scientific basis for public health decisions.

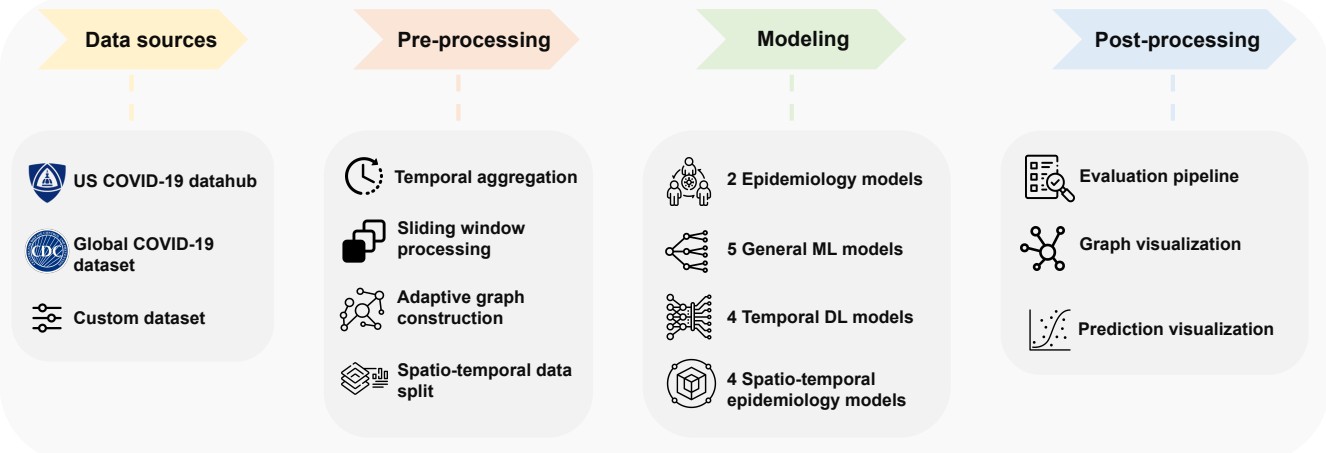

Figure 1: Structure and pipeline of using DeepEST.

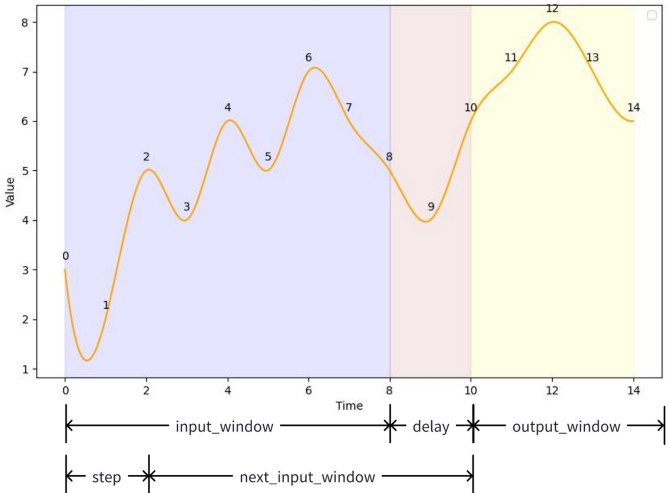

Figure 2: Sliding window processing example, illustrating the method to choosing input window and output window for time series forecasting. The data sequence in this graph is divided into three segments: window, delay, and horizon. The window (blue) represents historical data for training, the delay (pink) is the gap ignored in prediction, and the horizon (green) is the future period to forecast. The stride indicates the step size for moving the window forward to create overlapping windows for model training.

- **Temporal data prediction models**: For situations that require in-depth analysis of disease development in a specific region, we provide classical time series models. These models are designed to help public health experts independently analyze and predict the development of disease in a given region using traditional and classical time series analysis methods. These models include some traditional epidemiological models, machine learning(ML)

models, and temporal deep learning models, which provide accurate predictions of disease development trends.

All of the four types of models are shown in Table 2.

Table 2: Supported models in DeepEST.

| Deep learning-based | | Machine learning-based | |
|---|---|---|---|
| Temporal | Spatio-temporal | General ML | Epidemiology |
| RNN [11] | STAN [13] | ARIMA [2] | SIR [1] |
| GRU [9] | ColaGNN [8] | XGBoost [4] | SEIR [16] |
| LSTM [17] | HOIST [12] | LightGBM [15] | |
| Transformer [19] | TAP-RSV [20] | SVM [6] | |
| | | Random Forest [3] | |

## 3.3 Modeling and Post-processing

DeepEST provides an efficient, automated process for building, training, and testing spatio-temporal and time series models. The module is designed to simplify the model development process and ensure consistency and repeatability of the analysis.

This module offers a unified model constructor model_builder, a trainer, and a tester for both time series models and spatio-temporal sequence models. These components provide users with a one-stop solution for configuring and training all models within the toolkit in a similar manner.

*3.3.1 Model construction.* We use the Model Builder tool to allow users to build all the models they need through a consistent process. Whether it is a complex spatio-temporal model or a classical temporal model, Model Builder can provide the necessary support to ensure the standardization and efficiency of model construction.

*3.3.2 Training and evaluation.* The training test module contains two key components, Trainer and Tester.

- The Trainer is responsible for training the model on the training set using historical data, adjusting the model parameters by optimizing the algorithm, and saving the model parameters with the best performance in the validation set.

- The Tester is used to evaluate the model's performance on a separate set of tests, verifying its prediction accuracy and generalization ability.

The trainer and tester can be divided into ml_trainer, ml_tester (for SIR, SEIR, Arima, XGBoost) and dl_trainer,dl_tester (for other models) according to the type of invoked model.

*3.3.3 Visualization tools.* We developed visualization tools to improve the efficiency of visualizing prediction results. This module can not only visualize the graph structure of the underlying data, but also show the prediction results of the model, such as the trend of disease development and transmission patterns (as shown in Figure 3) and Figure 4. This visualization capability allows non-technical users to easily understand the output and predictions of the model.

The design of the training and test module focuses on user-friendliness and modularity, so that even users without deep technical background can easily get started, while ensuring the flexibility and scalability of the module to adapt to the development of future research needs.

## 4 THE PROCESS OF USING DEEPEST

The primary target audience for DeepEST consists of healthcare professionals, public health policymakers, and deep learning researchers. Some of these users typically do not possess an extensive background in data science. However, they require the utilization of AI technology for disease trend analysis and forecasting to support their decision-making processes in their professional roles. DeepEST aims to simplify the use of models through streamlined class calls, reducing the difficulty for users.

Initially, the process begins with the input of both static and dynamic data. Following this, the user should specify the input window, output window, and the normalization method to be employed. These parameters are critical for preparing the data appropriately for model training.

Subsequently, the user is required to determine whether a spatio-temporal (ST) model is to be employed. If an ST model is selected, the subsequent step involves constructing a graph to encapsulate the spatial and temporal dependencies inherent in the data. Conversely, the process can proceed directly to the subsequent steps without the necessity of graph construction.

Once the preprocessing and model-type decisions are finalized, the next step is to build the model. This involves setting up the model architecture and configuring the hyperparameters according to the specific requirements of the task. The model is then trained using the training dataset and validated against a separate validation dataset to fine-tune its parameters.

After the training phase, the model undergoes testing with an independent test dataset to evaluate its performance and generalization capability. The final phase of the process is visualization, where the results of the model are graphically represented to facilitate interpretation and analysis. This step is essential for understanding the model's predictions and assessing its efficacy.

In Figure 5, we provide an example code snippet to show how DeepEST processes the spatio-temporal data, trains a state-of-the-art deep learning model and evaluates the results with 10 lines of codes.

## 5 CONCLUSIONS

The development of DeepEST marks a significant advancement in the field of spatio-temporal epidemiology prediction. By harnessing the power of deep learning and integrating it seamlessly with traditional epidemiological models, DeepEST offers an unprecedented tool that empowers public health researchers with limited technical expertise to engage in complex epidemiological forecasting. This library addresses a crucial gap in the availability of accessible, advanced predictive modeling tools, making it easier to analyze complex spatial and temporal patterns in disease spread. Through its comprehensive model zoo and intuitive tools for data preprocessing, evaluation, and visualization, DeepEST facilitates a deeper understanding of disease dynamics and enhances the ability of public health officials to make informed decisions during health crises.

As the global health landscape continues to evolve, tools like DeepEST are essential for anticipating and mitigating the impacts of infectious diseases. The ongoing development and refinement of DeepEST will focus on incorporating more diverse data sources, improving model accessibility, and expanding its applicability to a wider range. This work contributes to the broader goal of equipping public health systems with the necessary tools to respond more effectively to both current and future epidemiological challenges.

## 6 DISCUSSION ON ALIGNING DEEPEST WITH AIDSH WORKSHOP SCOPE

The "Artificial Intelligence and Data Science for Healthcare (AIDSH)" workshop is dedicated to exploring the convergence of Artificial Intelligence (AI) and Data Science (DS) within the healthcare sector. It places a significant emphasis on bridging the gap between data-centric AI and people-centric healthcare (PCHC). The workshop aims to foster innovative applications of AI and data science in healthcare while encouraging active participation from individuals in managing health data (including electronic health records and public health data).

Our paper, DeepEST, aligns with the AIDSH workshop's research scope in several key aspects:

- **AI Applications on Epidemiology Prediction**: DeepEST is a Python library specifically designed for spatio-temporal epidemiology prediction. It employs deep learning techniques to analyze patterns of disease spread over time and across different regions, providing accurate forecasts for public health crises. This focus on prediction aligns with the workshop's goal of using AI and data science to drive innovation in healthcare.
- **Advancing People-Centric Healthcare (PCHC)**: The design of DeepEST aims to lower technical barriers, allowing more individuals to engage in epidemiological research and forecasting. It emphasizes simplifying the process of model development and application, making it accessible to users without extensive technical backgrounds. This user-friendliness is conducive to enhancing interactivity, encouraging users (researchers and public health experts) to actively participate in epidemiological forecasting and data analysis. This accessibility is in line with the PCHC philosophy.
- **Education Support**: The availability of our open-sourced code and the comprehensive suite of tools provided by DeepEST can

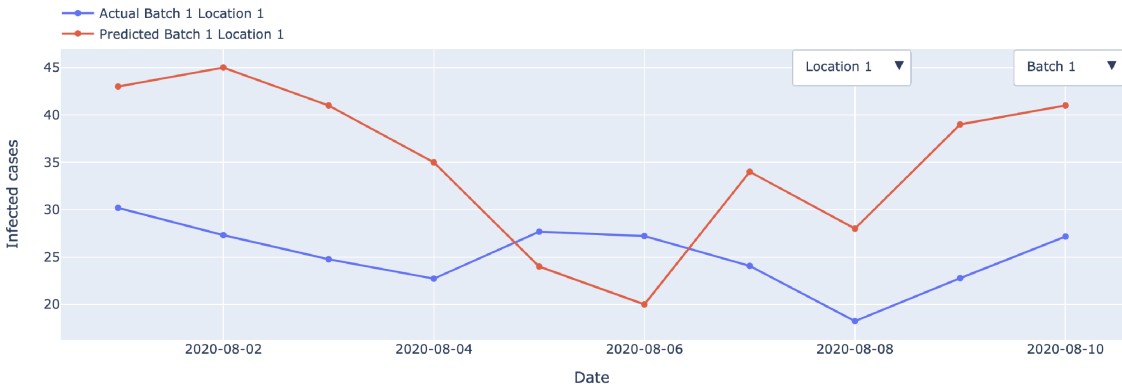

**Figure 3: Example visualization of comparison between prediction data and true data from 2020-08-01 to 2020-08-10. The actual cases are represented by a blue line, while the predicted cases are depicted with a red line. The graph shows fluctuations in the number of cases over time, illustrating discrepancies between the actual data and the predicted model. Dropdown selectors for location and batch indicate the graph's capability to display data for different scenarios.**

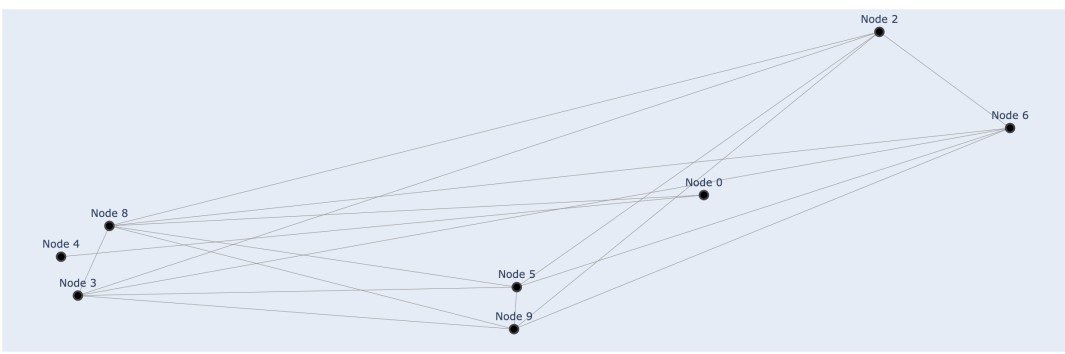

**Figure 4: Example visualization of graph, which represents relationships or interactions among different location. Nodes are labeled numerically, representing different locations, and are interconnected with lines.**

serve as a foundation for educational initiatives, helping public health officials and researchers learn to apply AI and DS in epidemiological forecasting. This aligns with the workshop's call for data education and support to empower individuals in healthcare.

In summary, DeepEST not only contributes to the field of spatio-temporal epidemiology prediction but also fits squarely within the research scope of the AIDSH workshop. It represents an effort to bridge the gap between data-centric AI and people-centric healthcare by providing a powerful yet accessible tool for predicting and managing public health crises data.

## ACKNOWLEDGMENTS

This work was supported by the National Natural Science Foundation of China (U23A20468), and Xuzhou Scientific Technological Projects (KC23143). Junyi Gao acknowledges the receipt of studentship awards from the Health Data Research UK-The Alan Turing Institute Wellcome PhD Programme in Health Data Science (grant 218529/Z/19/Z).

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

```
>>> from deepest import dl_trainer, dl_tester, build_model, preprocess_data, construct_adjacency_matrix
# Data was preprocessed with z-score normalization, setting input and output windows to 7 and 3
respectively, and partitioned into training, validation, and test sets in a 3:1:1 ratio in temporal
dimension.
>>> st_data = preprocess_data(dynamic_data, static_data, label, input_window=7, output_window=3,
normalization="z-score", type="spatial-temporal", temporal_ratio =[0.6,0.2,0.2])
# Get the data_loader with batch_size as 64
>>> trainset, valset, testset = st_data.get_dl_dataset()
>>> train_st_loader, val_st_loader, test_st_loader = get_dataloader(trainset, batch_size=64,
shuffle=True), get_dataloader(valset, batch_size=64), get_dataloader(testset, batch_size=64)
# Calculate the Euclidean distance between static data of each node and filter nodes with a distance
less than 0.5 to construct a graph.
>>> edge_index = construct_adjacency_matrix(static_data, distance_matric="euclidean", threshold=0.5)
# Construct the model by preprocessed data and model name.
>>> model = build_model(preprocess=st_data, model_name="colagnn", edge_index=edge_index)
# build trainer and tester from model
>>> trainer = dl_trainer(model= model)
>>> tester = dl_tester(model=model)
# train the model for 10 epochs
>>> trainer.train(train_dataloader= train_st_loader, val_dataloader= val_st_loader, epochs= 10)
# test the model, and get the MSE between true data and predict data
>>> tester.test(test_st_loader, metric="MSE")
```

**Figure 5: Example code of using DeepEST to build a spatio-temporal prediction model.**

(2020), 533–534.

[11] Jeffrey L Elman. 1990. Finding structure in time. *Cognitive science* 14, 2 (1990), 179–211.

[12] Junyi Gao, Joerg Heintz, Christina Mack, Lucas Glass, Adam Cross, and Jimeng Sun. 2023. Evidence-driven spatiotemporal COVID-19 hospitalization prediction with Ising dynamics. *Nature communications* 14, 1 (2023), 3093.

[13] Junyi Gao, Rakshith Sharma, Cheng Qian, Lucas M Glass, Jeffrey Spaeder, Justin Romberg, Jimeng Sun, and Cao Xiao. 2021. STAN: spatio-temporal attention network for pandemic prediction using real-world evidence. *Journal of the American Medical Informatics Association* 28, 4 (2021), 733–743.

[14] Melissa A Haendel, Christopher G Chute, Tellen D Bennett, David A Eichmann, Justin Guinney, Warren A Kibbe, Philip RO Payne, Emily R Pfaff, Peter N Robinson, Joel H Saltz, et al. 2021. The National COVID Cohort Collaborative (N3C): rationale, design, infrastructure, and deployment. *Journal of the American Medical Informatics Association* 28, 3 (2021), 427–443.

[15] Guolin Ke, Qi Meng, Thomas Finley, Taifeng Wang, Wei Chen, Weidong Ma, Qiwei Ye, and Tie-Yan Liu. 2017. Lightgbm: A highly efficient gradient boosting decision tree. *Advances in neural information processing systems* 30 (2017).

[16] William Ogilvy Kermack and Anderson G McKendrick. 1927. A contribution to the mathematical theory of epidemics. *Proceedings of the royal society of london. Series A, Containing papers of a mathematical and physical character* 115, 772 (1927), 700–721.

[17] Pankaj Malhotra, Lovekesh Vig, Gautam Shroff, Puneet Agarwal, et al. 2015. Long Short Term Memory Networks for Anomaly Detection in Time Series.. In *Esann*, Vol. 2015. 89.

[18] Benedek Rozemberczki, Paul Scherer, Yixuan He, George Panagopoulos, Alexander Riedel, Maria Astefanoaei, Oliver Kiss, Ferenc Beres, Guzman Lopez, Nicolas Collignon, et al. 2021. Pytorch geometric temporal: Spatiotemporal signal processing with neural machine learning models. In *Proceedings of the 30th ACM international conference on information & knowledge management*. 4564–4573.

[19] Ashish Vaswani, Noam Shazeer, Niki Parmar, Jakob Uszkoreit, Llion Jones, Aidan N Gomez, Łukasz Kaiser, and Illia Polosukhin. 2017. Attention is all you need. *Advances in neural information processing systems* 30 (2017).

[20] Chaoqi Yang, Junyi Gao, Lucas Glass, Adam Cross, and Jimeng Sun. 2024. Multifaceted analysis and prediction for the outbreak of pediatric respiratory syncytial virus. *Journal of the American Medical Informatics Association* 31, 1 (2024), 198–208.

## A APPENDIX: FUTURE WORK

DeepEST is still in the developing phase. We list the future development plan before the first formal release:

(1) **Improving the encapsulation and abstraction of the functions and classes**: We will refine the encapsulation of model and data classes to promote better code reuse and maintainability. We will also provide a set of well-packaged basic modules, including common graph neural networks and recurrent neural networks, to facilitate the construction of custom models by users. These modules will be designed for ease of integration and high performance.

(2) **Adding pre-processed regional demographics and census data**: We will integrate census data and enhance the range of static features available within the package. Users will be able to query data using various geographic identifiers such as zip code, ISO, and FIPS codes, making it easier to incorporate demographic information into analyses.

(3) **Adding post-analysis tools**: We will develop and incorporate tools for post-analysis, including the calculation of prediction confidence intervals and measures of uncertainty. These tools will enable users to better assess the reliability and robustness of their model predictions.

(4) **Adding more visualization tools**: We will implement advanced visualization tools, including map visualizations, to help users better understand and interpret their data. These visualizations will be interactive and customizable, allowing for detailed exploration of spatial and temporal patterns.

(5) **Adding more models and dataset parser**: We will expand the library with additional models to cover a wider range of applications and use cases. This will include integrating state-of-the-art models from the latest research. We also aim to develop

and include more dataset parsers to support a broader array of data formats and sources, making it easier for users to import and work with their data within the `DeepEST` framework.

(6) **Adding support for R version**: As many public health researchers prefer the R language, we aim to develop and maintain an R version of the `DeepEST` package, ensuring that the functionalities are accessible to users who prefer or require R for their data analysis workflows.

(7) **Providing video and notebook tutorials**: We also aim to create comprehensive video tutorials covering various aspects of the package, from installation and setup to advanced usage and customization. These videos will serve as a visual guide to help users understand and utilize the package effectively. We will develop Jupyter notebook tutorials that include step-by-step instructions and examples. These notebooks will provide hands-on experience with the package's features and functionalities, enabling users to follow along and practice on their own datasets.