# OpenReview forum: "DeepEST: A Python Library for Spatio-Temporal Epidemiology Prediction"
_KDD.org/2024/Workshop/AIDSH — KDD-AIDSH 2024 Poster_

### Official Review · Reviewer_K1A7 · 2024-06-19

**Rating:** 7
**Confidence:** 3

**Review:**

Summary: The paper introduces DeepEST, a Python library tailored for spatio-temporal epidemiology prediction to simplify the utilization of models for disease trend analysis and forecasting. DeepEST streamlines the process by guiding users through data input, preprocessing, model selection, training, validation, testing, and visualization. By integrating deep learning techniques with traditional epidemiological models, DeepEST empowers public health researchers with limited technical backgrounds to engage in complex epidemiological forecasting. The library bridges the gap between data-centric AI and people-centric healthcare, aligning with the goals of the AIDSH workshop. DeepEST democratizes access to predictive modeling tools, making them more accessible and user-friendly for public health experts and researchers.

Strengths:
1. The paper is well-written and organized.
2. DeepEST reduces technical barriers and allows more individuals to engage in epidemiological research and forecasting.
3. The open-sourced code provided has the potential usage as a foundation for educational initiatives.

Weaknesses:
The language has better to be polished before publication.

---

### Official Review · Reviewer_R9Ay · 2024-06-20

**Rating:** 6
**Confidence:** 4

**Review:**

### Brief Summary

The paper introduces DeepEST, a Python library designed to facilitate deep learning-based spatio-temporal epidemiology prediction. It aims to address the gap in accessible predictive modeling tools for public health researchers. DeepEST integrates advanced predictive modeling techniques such as graph neural networks (GNNs) and recurrent neural networks (RNNs) with traditional epidemiological models. The library offers tools for data preprocessing, model development, and visualization, simplifying the process of model application and reducing technical barriers for researchers.

### Strengths
- Relevance to Current Needs: Given the ongoing and future public health challenges, the development of such a tool is highly relevant and timely.
- Open-Source Availability: The availability of the code supports transparency, collaboration, and further development by the community.

### Weaknesses
- Lack of usage information in README.md: for example, there seems to have no information regarding data downloading and pre-processing, and there is no specification on how to train and evaluate different models
- Lack of including advanced models in the literature: The COVID-19 Hub, https://github.com/reichlab/covid19-forecast-hub/tree/master/data-processed, has included many specifically designed models for epidemiological forecasting, and some of them also leveraged the spatial-temporal information and have been open sourced. These models should be included into this library.
- Lack of experimental results comparing different models: This work, at the current stage, does not include extensive results of all included models, leading to insufficient insights delivered.

---

### Decision · Program_Chairs · 2024-06-28

Accept (Poster)